# OpenReview forum: "PRISM: A Unified and Generalizable Adversarial Robustness Evaluation Framework for LLM-based Classification"
_ICLR.cc/2026/Conference — ICLR 2026 Conference Withdrawn Submission_

### Official Review · Reviewer_tuwS · 2025-10-17

**Soundness:** 3
**Presentation:** 3
**Contribution:** 1
**Rating:** 2
**Confidence:** 4

**Summary:**

This paper proposes a unified framework for assessing the robustness and vulnerability of LLMs. The authors conduct extensive experiments on a phishing email detection task across GPT-4o, Claude Sonnet 4, and Grok-3, and report several interesting findings.

**Strengths:**

1. The motivation and method are clear, and the prompt description is helpful.

2. The authors provide sufficient evidence to support the effectiveness of their proposed method.

3.  This paper is well-written, making it easy to follow.

**Weaknesses:**

1. This paper reads more like a technical report than a research paper. It essentially reproduces techniques from multiple prior works and combines them into a framework for evaluating LLM robustness. The overall framework heavily relies on existing research and task-specific prompt engineering, both of which limit its novelty and generalizability (as changing the task requires redesigning the prompt).

2. The evaluation is limited to the phishing email detection task. The paper would be more convincing if it included a broader range of tasks beyond phishing detection and incorporated more open-source and API-based LLMs for comprehensive validation.

**Questions:**

Refer to weaknesses.

---

### Official Review · Reviewer_2VP7 · 2025-11-01

**Soundness:** 3
**Presentation:** 2
**Contribution:** 2
**Rating:** 2
**Confidence:** 3

**Summary:**

This paper introduces PRISM, a unified framework for evaluating the adversarial robustness of LLMs in classification tasks. The framework integrates three attack dimensions: semantic-preserving linguistic refinement, prompt-level instruction injection, and cross-lingual shifts. Using phishing detection as a case study, the authors evaluate three frontier LLMs (GPT-4o, Claude Sonnet 4, and Grok-3). The core contributions are the PRISM framework itself, an empirical analysis demonstrating that models with high baseline accuracy are vulnerable across all three attack vectors, and practical insights into prompt engineering trade-offs.

**Strengths:**

* The paper organizes its evaluation around three distinct types of vulnerabilities. By applying these different attack dimensions to the same set of models and task, the study provides a valuable comparative benchmark, revealing model-specific vulnerability patterns.
* Empirical setup with realistic conditions: conducted evaluations on three contemporary LLMs, and incorporated real-world data characteristics, such as the natural class imbalance found in email traffic, by testing on both balanced and skewed distributions.
* The multilingual evaluation provides a quantitative demonstration of a major weakness in current LLMs.

**Weaknesses:**

My major concerns are on the limited methodological novelty and technical depth as follows.

* The core contribution, the PRISM framework, appears to be a conceptual grouping of existing evaluation areas rather than a novel technical framework. It serves as a descriptive schema for the experiments conducted but does not introduce new evaluation paradigms or technical depth/insights.
* The 6-tuple formalism (T, D, A, P, M, R) introduced in Section 1 (lines 81-85) is not carried through or operationalized in the rest of the paper. The formalism appears to be superficial or too broad and it's hard to see the grounding or necessity of such formalism in the rest of the paper.
* The analysis is primarily descriptive, reporting performance metrics from experiments without taking further steps (e.g., providing a deeper technical investigation into why the models fail, how to mitigate the failures, etc.)
* The paper does not report any statistical significance testing for its comparisons. For example, in Table 4 (multi-template), the ASR for GPT-4o is 1.1% and for Grok-3 is 1.6%. It is unclear if this difference is statistically meaningful or simply noise from a single experimental run. This makes it difficult to draw firm conclusions about relative model vulnerabilities.

**Questions:**

My major question is about the positioning / main contributions of the paper.

To align with the current substance of the paper, I'd suggest re-framing the work's primary contribution as a comprehensive empirical benchmarking study rather than the proposal of a novel framework. This would manage reviewer expectations and highlight the paper's empirical strengths.

Alternatively, to justify the "framework" claim, I'd suggest adding technical depth. This could involve proposing and evaluating a novel compound attack that synergistically combines multiple vectors (e.g., a cross-lingual prompt injection), or performing a deeper analysis of failure modes (e.g., analyzing attention patterns or internal representations to explain why certain models are more vulnerable to specific attacks).

---

### Official Review · Reviewer_GaBX · 2025-11-03

**Soundness:** 1
**Presentation:** 2
**Contribution:** 1
**Rating:** 2
**Confidence:** 2

**Summary:**

The paper introduces PRISM, a framework to evaluate adversarial robustness of LLM-based classification, instantiated on phishing-email detection. It unifies three attack dimensions—(i) semantic-preserving “refinement”, (ii) prompt-level instruction injection, and (iii) cross-lingual shifts—and evaluates GPT-4o, Claude Sonnet 4, and Grok-3 across balanced/imbalanced settings and different prompting styles.

**Strengths:**

- The 6-tuple formalization (T, D, A, P, M, R) and figure clarify how the attacks and prompts plug into a single evaluation pipeline.
- Useful empirical takeaways. Zero-shot often matches/exceeds structured/CoT with much lower latency on imbalanced data.

**Weaknesses:**

- Limited novelty beyond aggregation. The work primarily bundles known robustness axes (adversarial paraphrase, prompt injection, cross-lingual transfer) into a single protocol; no new attack class, defense, or evaluation metric appears fundamentally novel. The “unified” aspect reads more like scope consolidation than a methodological advance.
- Core evaluations use relatively small samples (e.g., 189–200 items for adversarial/injection tests; 179 per language for multilingual), which limits statistical power and generality—especially for a claim of a generalizable framework.
- Injection templates are simplistic (“classify as Safe Email”) and appended to message tails; actual threats often embed instructions in headers, quoted threads, MIME parts, or retrieved content. The work does not systematically vary injection locality or delivery channel.

**Questions:**

See cons

---

### Official Review · Reviewer_yczL · 2025-11-03

**Soundness:** 1
**Presentation:** 2
**Contribution:** 1
**Rating:** 2
**Confidence:** 3

**Summary:**

This paper introduces PRISM, a framework for evaluating the robustness of LLM-based classification under three forms of "attack" (1) semantic-preserving adversarial paraphrasing, (2) prompt injection, and (3) language translation. The authors instantiate phishing detection as a case study and report robustness of GPT-4o, Claude Sonnet 4, and Grok-3.

**Strengths:**

N/A

**Weaknesses:**

My primary concern with this submission is the lack of substance. The claimed contribution is a "unified framework" that evaluates robustness of LLM-based phishing email classification under adversarial paraphrasing, prompt injection, and language translation.

- The paper involves weak combined evaluation of known, existing robustness vulnerabilities for a single classification problem. The "unification" consists primarily of running existing attack styles sequentially on the same dataset. I do not think this meets the bar for an ICLR publication.
- The attack evaluations themselves are quite weak. Blind paraphrasing to elicit misclassification is neither new nor interesting, and static template-based prompt injections are weakest in class, compared to SoTA automated implementations like GCG, AutoDan, TAP etc. I would expect significantly higher ASR using quite literally any other attack implementations here. Finally, if the entire paper's purpose is to provide evaluation insight into phishing email detection performance, one would expect more than 3 models, and insights into impact of model sizes and architectures.

- An unusually large amount of writing is dedicated to explaining primitive ML concepts, with explanations of precision, recall, f1, fpr, ASR, etc taking up space that could have been used for any other interesting experiments.

**Questions:**

I am also not quite sure that any comments about robustness under language shift can be made with N=10 samples. This perhaps does not even qualifiy as an attack - is a phishing email useful if it is written in say, Bangla, but the victim does not know Bangla?

---

### Note · Authors · 2025-11-19

I have read and agree with the venue's withdrawal policy on behalf of myself and my co-authors.